

# QCD or what?

Theo Heimel[1], Gregor Kasieczka[2], Tilman Plehn[1*] and Jennifer M. Thompson[1]

**1** Institut für Theoretische Physik, Universität Heidelberg, Germany
**2** Institut für Experimentalphysik, Universität Hamburg, Germany

⋆ plehn@uni-heidelberg.de

## Abstract

Autoencoder networks, trained only on QCD jets, can be used to search for anomalies in jet-substructure. We show how, based either on images or on 4-vectors, they identify jets from decays of arbitrary heavy resonances. To control the backgrounds and the underlying systematics we can de-correlate the jet mass using an adversarial network. Such an adversarial autoencoder allows for a general and at the same time easily controllable search for new physics. Ideally, it can be trained and applied to data in the same phase space region, allowing us to efficiently search for new physics using un-supervised learning.



**Content**

# 1 Introduction

Since the start of the LHC, jets have turned from an experimental annoyance to the most interesting and powerful analysis objects. Together with a vastly improved understanding of multi-jet kinematics, we have learned how to use jet constituents to identify LHC signals [1–3]. This way, jets no longer serve as universal analysis objects, but merely separate jet-level observables from subjet observables [4–9]. A second development in LHC analyses is that we compare simulated and observed jet events at the detector level, instead of first-principles theory and data [10–15]. This raises the question why we still rely on intermediate high-level observables rather than low-level observables like 4-momenta of particle flow objects. The latter are driving deep learning applications at the LHC, where the term deep learning really describes the shift from high-level to low-level input observables [16–31]. In this paper we show how this new approach allows us to tackle the basic question:

*Do our observed jets really look like QCD jets?*

In addition to testing well-defined hypotheses, neural networks also allow us to search for anomalies without ever defining a signal. We can just study QCD jets in data and use machine learning techniques to search for non-QCD patterns. The appropriate network architecture are autoencoders [32–37], networks which compress their information to search for patterns which are no longer described by the compressed representation.

Once we abandon high-level subjet observables we can choose our input format to deep learning analysis tools. This allows us to pick a data format that is best suited to a given problem. The most frequently used format are jet images, calorimeter entries in the azimuthal angle vs rapidity plane, analyzed through image recognition [38]. They can be used to identify hadronic decays of weak bosons [16–18, 25, 26] or top quarks [27, 28], or to distinguish quark-like from gluon-like jets [30,31]. A limiting factor for jet images are measurements with vastly different angular resolution, like calorimeter and tracker measurement [29–31]. For example in this case we can directly use 4-momenta [19–24] and, inspired by graph convolutional networks, analyze them efficiently using the Minkowski metric [29]. These approaches can also be generalized to search for new physics at the event level [39–43]. For all network setups we can visualize their behavior based on truth-level information in Monte Carlo simulations [27, 30, 31, 44].

Deep learning applications to jet physics at the LHC face three key limitations:

1. the availability of training data;
2. systematic uncertainties in our understanding of the training data;
3. control over the exact physics question which the network answers.

While most available studies answer well-defined physics questions based on labelled or simulation data, the first limitation can be tackled by switching to weakly supervised learning [45–49]. In this paper we will go even further and show how autoencoders work in the absence of a signal sample. The second challenge can for example be addressed with adversarial networks, de-correlating for example kinematic information or theory assumptions [50–57]. Alternatively, refiner networks [84] can be used to improve the quality of simulation. For our autoencoder we will de-correlate kinematic information like the jet mass, generating experimental control regions and controlling systematics related to the composition of the training sample. This way, the autoencoder can be trained and applied on the same phase space and avoid systematic uncertainties from relating simulation to data or background to signal phase space regions. The same de-correlation technique also allows us to tackle the third challenge. A promising approach in this direction is to combine well-understood features like mass peaks

with implicit, orthogonal information [58]. More generally, we will show how any well-defined physics effect can be de-correlated from the autoencoder analysis, allowing us to construct control regions, side bands, or a flat or smoothly falling spectrum suitable for a bump hunt in any observable needed for a given analysis.

A set of events flagged by the autoencoder as anomalies does not automatically qualify as a signal of new physics. It is standard experimental procedure to test whether any signal could be caused by detector effects. Typical tests include checks whether events cluster geometrically (all jets originate from a specific region in the $\eta - \phi$ plane, hinting at a misbehaving region of the calorimeter) or temporally (from a specific run or run-period, hinting at problematic LHC or detector conditions). In the case of autoencoding jet images, an additional test would be an analysis of the correlation with well-understood substructure variables such as n-subjettiness, which is opportune to understand the topology of the identified signal. Finally, mis-calibrations of the jet-energy that cause an artificial mass peak can be taken care of using control regions — if the mass peak is present in sidebands as well, it is likely a miscalibration. All of these are relevant experimental considerations and should be included in any concrete study. However, autoencoding is no more susceptible (and arguably less so) than traditional techniques based on MC simulation.

In Sec.2.1 we will start by constructing an autoencoder [32–37] based on a convolutional network [59], in our case the image-based DEEPTOP tagger [27, 28]. Alternatively, we can analyze 4-vectors in an autoencoder version of the DEEPTOPLOLA tagger, as shown in Sec. 2.2. Next, we will control what kind of information the network uses by taking out the jet mass distribution through an adversarial network in Sec. 2.3. This allows us to devise a convincing side band analysis on the jet mass for the anomaly search [58]. With the help of these sidebands we can study the stability of the autoencoder network trained on not fully controlled, impure QCD samples in Sec. 2.4. After establishing our new methods using top tagging we will test them on scalar decays to four jets in Sec. 3.1 and on non-QCD showers in Sec. 3.2.

## 2 Autoencoded QCD vs tops

The aim of our study is to identify jets with an exotic, non-QCD origin using a neural network that is only trained on QCD jets. This can be done with autoencoder networks, which are stacks of networks layers with an intermediate set of bottleneck layers with a strongly reduced number of units, corresponding to a latent space with reduced dimensionality. Such a bottleneck can be added to convolutional networks [59], but it can also be added to a LOLA-like network working on constituent 4-vectors. The main structural change is that autoencoders do not work towards an output value which, assuming the right loss function, gives a probability for a jet being either signal or background. Instead, the network on both sides of the bottleneck is approximately symmetric, and the loss function is the difference between the input and the output. Once we run such a trained network on a test sample the loss function will tell us how well the network with its bottleneck encodes the features of the test sample.

An established, albeit non-glamorous benchmark for subjet studies are boosted hadronic top decays. This is why we first test our new autoencoder setup, trained on QCD jets, for anomalous top jets. After we benchmark autoencoders for image-based and 4-vector-based architectures, we will introduce a de-correlation with the jet mass. This approach can be immediately generalized to any other variable, defining plenty of control regions and side bands to control the autoencoder in an experimental reality.

## 2.1 Jet images

As long as we focus on the pixellated energy, we can analyze jets using a convolutional neural network (CNN) to learn jet images. Our autoencoder architecture is based on the DEEPTOP tagger [27], with significant improvements especially to the image pre-processing, developed in Ref. [28].

Our top and QCD samples are similar to the sample used in our DEEPTOP studies [27, 29]. We simulate top pair and di-jet production with PYTHIA8.2.15 [60] and DELPHES3.3.2 [61] for a collider energy of 14 TeV. For the QCD sample we do not distinguish between hard quarks and gluons. While the simulation used for these studies did not include effects of multi-parton interaction and pile-up, this is not a fundamental limitation of the proposed approach. Autoencoders can also be applied to the constituents of a jet after applying standard experimental techniques for the removal of pile-up [62–65]. A combination with grooming algorithms is possible as well, but would potentially limit the sensitivity as grooming makes explicit assumptions on how a shower ought to unfold.

Similarly, no detailed detector simulation was included. We expect the autoencoder to learn novel jet-shape variables from the distributions of constituents. There is no a-priori reason why these jet shapes would suffer from larger effects due to the detector simulation than widely used variables like groomed mass, n-subjettiness or energy correlation functions. For the practical application of the autoencoder we foresee training on data, making this technique even less subject to differences between data and simulation than ordinary approaches.

The substructure containers are fat anti-$k_T$ jets [66] with distance parameter $R = 0.8$, defined by FASTJET3.1.3 [67, 68]. They are required to have a transverse momentum in the range

$$p_{T,j} = 550 \dots 650 \text{ GeV} . \tag{1}$$

In addition they must be central, $|\eta_j| < 2$. For all signal jets we require both the truth-level partonic top and its decay products to be within the area of the fat jet. The inputs of the subjet analysis are particle flow objects [69] from the DELPHES E-flow. In the left panel of Fig. 1 we show the number of particle flow constituents for signal and background jets. The main feature is that already based on the larger number of constituents we could identify the hadronic top decays.

Following Ref. [28] we employ an improved pre-processing of the jet images, most notably applied before pixelization. This approach is directly motivated by the particle flow approach, which combines the coarse calorimeter information with the high-resolution tracker and provides us with a set of high-resolution 4-vectors. The center of the image is not defined by the hardest object, but by the $k_T$-weighted centroid of the fat jet constituents. The major principle axis is then turned to 12 o'clock. Finally, the image is flipped along the $x$-axis and $y$-axis, to ensure that the hardest constituent is located in the upper right quadrant. Only after this pre-processing we pixelize the images into a $40 \times 40$-pixel image, covering $\eta = -0.57 \dots 0.57$ and $\phi = -0.69 \dots 0.69$ around the center of the fat jet. The entries of the calorimeter images are given by the transverse momentum entering the detector cell, *i.e.* the sum of the transverse momenta of all particle flow objects covered by a pixel. Also in the left panel of Fig. 1 we show the number of non-zero pixels per image. The full image with its 1600 pixels is indeed sparsely filled. Each of the pixels is finally normalized to the sum of all pixels in the jet image. These images define the input and the output format of the autoencoder network.

The architecture of the autoencoder network is shown in Fig. 2. We use KERAS [70] combined with TENSORFLOW [71] to build our network. It is almost symmetric between the input

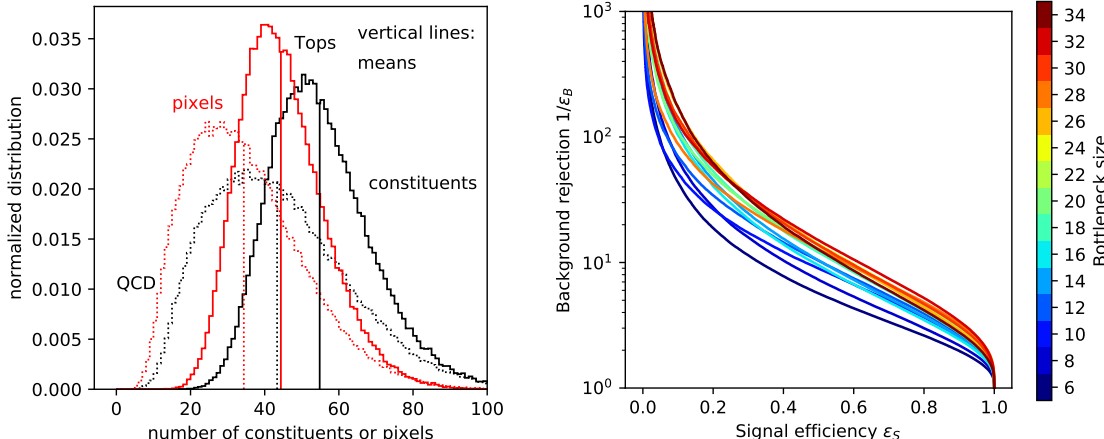

Figure 1: Left: numbers of constituents and of non-zero pixels for tops and QCD jets, 400,000 jets in total. Right: ROC curves for the image-based autoencoder identifying anomalous top jets for different bottleneck sizes.

and the output. The loss function is simply

$$L_{\text{auto}} = \sum_{1600 \text{ pixels}} \left( k_T^{\text{norm,in}} - k_T^{\text{auto}} \right)^2 \qquad (2)$$

in terms of the normalized input image and the autoencoder output image. We use the PReLU activation function throughout the network, to avoid a zero pseudo-solution, except for a linear activation function in the last layer. We use the ADAM optimizer [72] for training the network.

The autoencoder is trained on 100,000 QCD or background jets for up to 100 epochs and allow for an early stopping after ten epochs with stable loss. Our test sample consists of 200,000 top jets and 200,000 QCD jets. The large test sample allows for a study of the performance on several independent samples, confirming that our ROC curves are stable. For a variable cut in the loss function we can evaluate the composition of the signal-like jets in terms of true top and true QCD jets. These two fractions define a ROC curve, as shown in the right panel of Fig. 1. For these curves we vary the size of the bottleneck from 6 to 34 units in the smallest dense layer shown in Fig 2. We see a sizeable variation with the bottleneck size, developing a stable high-performance plateau between 20 and 34. It gives a stable area under curve (AUC) around 0.89 with a loss around $10^{-5}$ per pixel. The size of the bottleneck has to be compared with the initially 1600 pixels, of which 10 to 70 are non-zero, and which the CNN pools to 400 combined pixels. This large bottleneck size indicate that the image architecture is not perfectly adapted to encode the relevant QCD vs tops information in a small network layer.

The large size of the test sample allows us to evaluate our autoencoder on separate, statistically independent test samples. While the corresponding spread does not account for system-

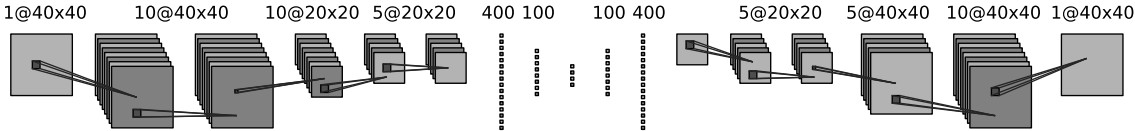

Figure 2: Architecture of the image-based autoencoder network. The $40 \times 40$ images are average-pooled to $20 \times 20$ images before entering the bottleneck. The dense units are first reduced from 400 to 100, the minimum size at the bottleneck is variable.

atics uncertainties related to the training, especially the training on data, it defines a statistical uncertainty of the autoencoder. It is shown as widths of the ROC curves, which are generated by evaluating the network on ten independent test samples with 20,000 QCD jets and 20,000 top jets each.

## 2.2 LoLa

When we want to include information beyond the calorimeter output, we can for example use the neural network based on the constituent 4-vectors developed for the DEEPTOPLOLA tagger [29]. It starts from a set of measured 4-vectors sorted by transverse momentum

$$(k_{\mu,i}) = \begin{pmatrix} k_{0,1} & k_{0,2} & \cdots & k_{0,N} \\ k_{1,1} & k_{1,2} & \cdots & k_{1,N} \\ k_{2,1} & k_{2,2} & \cdots & k_{2,N} \\ k_{3,1} & k_{3,2} & \cdots & k_{3,N} \end{pmatrix}. \tag{3}$$

Following the left panel of Fig. 1 we use $N = 40$ constituents, after checking that an increase to $N = 120$ does not make a measurable difference. For jets with fewer constituents we naturally fill the entries remaining in the soft regime with zeros.

To remove all information from the jet-level kinematics we boost all 4-momenta into the rest frame of the fat jet. This also improves the performance of our network. Inspired by recombination jet algorithms we can add linear combinations of these 4-vectors with a trainable matrix $C_{ij}$, defining a combination layer

$$k_{\mu,i} \xrightarrow{\text{CoLa}} \tilde{k}_{\mu,j} = k_{\mu,i} \, C_{ij} \qquad \text{with} \quad C = \begin{pmatrix} 1 & 1 & 0 & \cdots & 0 & C_{1,N+2} & \cdots & C_{1,M} \\ \vdots & 0 & 1 & & \vdots & C_{2,N+2} & \cdots & C_{2,M} \\ \vdots & \vdots & \vdots & \ddots & 0 & \vdots & & \vdots \\ 1 & 0 & 0 & \cdots & 1 & C_{N,N+2} & \cdots & C_{N,M} \end{pmatrix}. \tag{4}$$

We allow for $M = 10$ trainable linear combinations. These combined 4-vectors carry information on the hadronically decaying massive particles. In the original LOLA approach we map the momenta $\tilde{k}_j$ onto observable Lorentz scalars and related observables [29]. Because this mapping is not easily invertible we do not use it for the autoencoder. Instead, we extend the 4-vectors by another component containing the invariant mass,

$$\tilde{k}_j = \begin{pmatrix} \tilde{k}_{0,j} \\ \tilde{k}_{1,j} \\ \tilde{k}_{2,j} \\ \tilde{k}_{3,j} \end{pmatrix} \xrightarrow{\text{LoLa}} \begin{pmatrix} \tilde{k}_{0,j} \\ \tilde{k}_{1,j} \\ \tilde{k}_{2,j} \\ \tilde{k}_{3,j} \\ \sqrt{\tilde{k}_j^2} \end{pmatrix}. \tag{5}$$

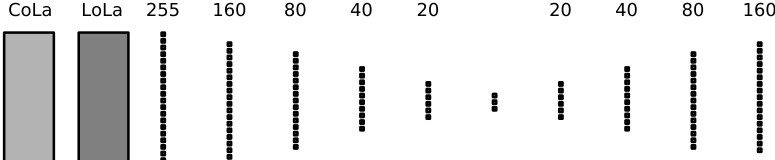

Figure 3: Architecture of the 4-vector-based autoencoder network. The 255 input units correspond to 55 LOLA-vectors with $4 + 1$ entries each. The output only consists of 160 units, because the extended 4-vectors only carry four independent observables.

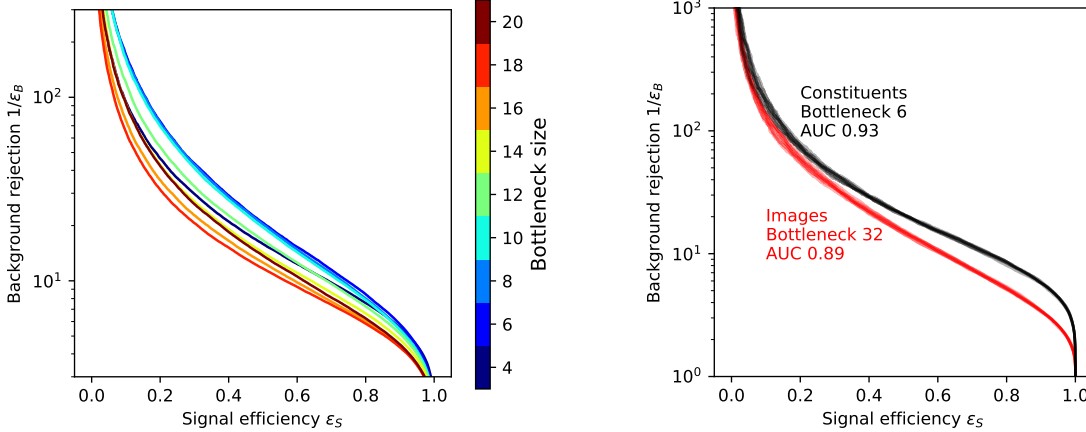

Figure 4: Left: ROC curves for the 4-vector-based or LoLa autoencoder identifying anomalous top jets for different bottleneck sizes. Right: comparison of the ROC curves for the image-based and the 4-vector-based autoencoders. The widths of the lines show the variation based on ten independent test samples for fixed training.

This defines a set of 51 extended 4-vectors, which form the input to our neural network. Again, we use KERAS [70] combined with TENSORFLOW [71]. Its architecture is shown in Fig. 3. The layer immediately after the LoLa contains $51 \times (4+1) = 255$ units. Between the second layer after LoLa and the last layer, the autoencoder network is symmetric. The final output consist of 40 4-vector-like objects, which can be compared with the corresponding second layer. The loss function is

$$L_{\text{auto}} = \sum_{j=1}^{40} \sum_{i=0}^{3} \left( \tilde{k}_{i,j}^{\text{in}} - \tilde{k}_{i,j}^{\text{auto}} \right)^2 \ . \tag{6}$$

As for the images we use the PReLU activation function, except for the last layer with its linear activation function, and the ADAM optimizer for the learning rate [72].

In the left panel of Fig. 4 we show the ROC curves for the 4-vector-based tagger for different choices of the bottleneck size. We now find the best result for a very small bottleneck with at most 10 units. The stable AUC value is around 0.92 with a loss around $10^{-5}$ per pixel. Such small functional bottlenecks reflect the fact that with the CoLa/LoLa structure we have encoded a lot of the relevant information in appropriate physics terms [29].

Finally, in the right panel of Fig. 4 we compare the best-performing image-based and 4-momentum-based autoencoders. The widths of the lines are again generated by evaluating the network on ten independent test samples. The main feature in this plot is that the LoLa-autoencoder does better than the image-based autoencoder. This is a result of the smaller possible bottleneck size, because the LoLa architecture is optimized to extract the leading discriminating features most efficiently. While this gives an advantage to the pure autoencoder, we will see the other side of the same medal in the next section.

## 2.3 De-correlating the mass

Neural networks separating signal and background jets after fully supervised training on labelled data are, in theory, straightforward to calibrate and understand. The problem at the LHC is that we hardly ever have enough labelled data to train such networks for relevant new physics searches — especially when the goal is to tag new resonances. Our autoencoder responds to this problem by limiting the training to QCD jets only and by only asking if a given

data set is described well by QCD or any other standard assumption. On the other hand, the more weakly the question is defined, the more important it is to control what the neural network actually learns. This is especially true when we use the network on low-level information rather then established high-level kinematic observables [73–79].

An established way to test a network is to exclude known, well-defined pieces of information from it through adversarial networks [51–57]. They consist of two networks playing against each other. Similar to generative adversarial networks, they can be used to train a network as an equivalent replacement for another data generator. In our application the additional adversary is trained to extract for instance the jet mass from the autoencoder output described in Eq.(2). In this image-based case a naive adversary loss function would read

$$L_{\mathrm{adv}}(M) = \left[ \widetilde{M} \left( \left| k_{T,i}^{\mathrm{adv}} - k_{T,i}^{\mathrm{auto}} \right| \right) - M \right]^2 , \tag{7}$$

with the inputs $k_{T,i}^{\mathrm{auto}}$, the outputs $k_{T,i}^{\mathrm{adv}}$, the given jet mass $M$, and the trained proxy to the jet mass $\widetilde{M}$. As we will discuss below, for our study we replace the exact function $\widetilde{M}$ with a binned determination of the jet mass [52]. The combined loss function which replaces Eq.(2) for the autoencoder can be written in terms of a Lagrangian multiplier [51, 52]

$$L = L_{\mathrm{auto}} - \lambda L_{\mathrm{adv}}(M) . \tag{8}$$

The Lagrangian multiplier $\lambda$ introduces a boundary condition, $L_{\mathrm{adv}} \to 0$, in case the adversary learns the mass perfectly. The value of $\lambda$ determines the balance between the two networks. While the task of the autoencoder network is to describe the QCD training data, the adversary extracts the jet mass from the autoencoder output. Playing against each other and minimizing the combined loss function with the relative sign, the combined network wants the adversary to be as unsuccessful as possible. The adversarial autoencoder will hence avoid all information on the jet mass or any other boundary condition. Note that at least for the top jets this only affects the fat jet mass and still leaves us with the $W$-mass in the clustering history.

As a starting point, we show the jet mass distribution after applying the image-based autoencoder. We know from many studies that the jet mass is a powerful observable in separating QCD jets from hadronically decaying heavy states. On the other hand, since we also know that a small fraction of QCD jets will feature large jet masses, we expect to see a top signal as a jet mass peak over a smooth QCD jet background.

In the left panel of Fig. 5 we show jet mass distributions for QCD jets in slices of the autoencoder loss function. The per-centile ranges from all QCD jets to the 5% least QCD-like of all QCD jets. For the full jet sample we see the expected peak at small $m_j \approx 50$ GeV with a long tail extending beyond 300 GeV. For the least QCD-like jets in the pure QCD sample a peak at $m_j \approx 200$ GeV appears. This means that the cut on the autoencoder output badly shapes the background and makes it signal-like. This defines the task of the adversarial network: provide a smooth jet mass distribution for QCD jets, independent of the value of the autoencoder loss function; or in other words, de-correlate the jet mass from the autoencoder.

Again, we use KERAS [70] and TENSORFLOW [71] with the ADAM [72] optimizer for the combined adversarial network. The image-based autoencoder part of the network is described in Fig. 2; the adversarial part consists of eight dense layers with 800, 400, 200, 100, 50, 25, 10, and 12 units. We now train this network on 600,000 QCD jets. The output layer corresponds to 10 pre-defined slices in the jet mass, binned such that they are populated by the same number of QCD jets. On each side we add overflow bins which are not populated by QCD jets. The task of the adversary is not to extract the exact jet mass value, but to determine the probabilities for the jet mass to fall into each bin. This statistical interpretation requires a multi-label cross

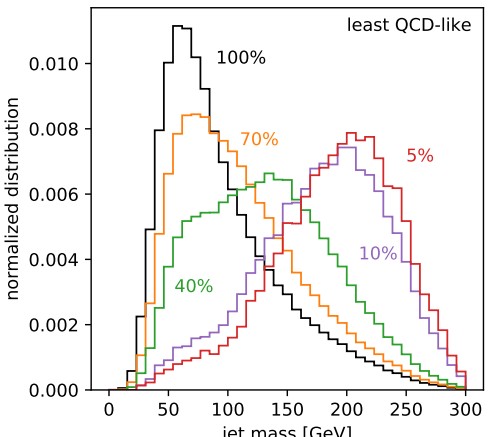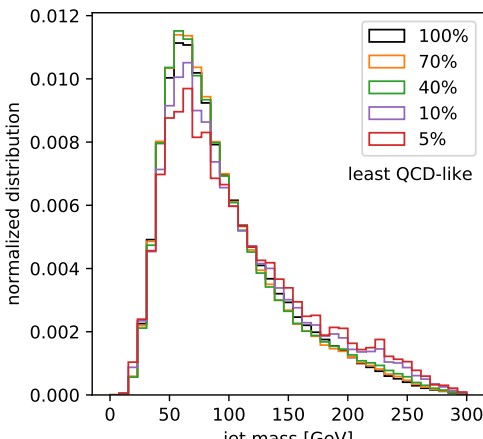

Figure 5: Left: jet mass distributions from the image-based autoencoder applied to QCD jets. The different lines show the full sample up to the 5% least QCD-like jets, defined by the autoencoder loss function. Right: the same jet mass distributions, but for the QCD-trained adversarial autoencoder network.

entropy as the adversary loss function [52]. All layers use the ReLU activation function except for the last layer, where a SoftMax activation function guarantees that all 12 probabilities sum to one. When training on the combined loss function, each epoch is split into batches of size 128. For each batch we first train the autoencoder using the combined loss function of Eq.(8) and then train the adversary with only the adversary loss function. The size of the Lagrangian multiplier is chosen such that the two contributions to the loss function are of similar size, *i.e.* it balances the de-correlation vs the discrimination power of the network. For instance, the jet mass distribution for $\lambda = 5 \cdot 10^{-4}$, shown in the right panel of Fig. 5, indicates that the background shaping is indeed largely gone.

To study the interplay of the mass de-correlation with the performance of the adversarial autoencoder we show results for three values of $\lambda$ in Fig. 6. For increasing values of $\lambda$ the background shaping indeed improves. On the other hand, we can illustrate the performance of the network by testing on QCD data with 3% top jets injected. For the full sample we indeed see a hint of top jets around $m_j = m_t$ in all three panels of Fig. 6. We can then extract the 5% least QCD-like jets, which should include most of the top jets. What we find is that the number of top jets in this selection is diluted from the maximum expected 3/5 of the 5% least QCD-like jets. This dilution grows with $\lambda$, because it is an effect of taking out the jet mass as the strongest discriminator from the network. The performance drop is given as AUC values and detailed in the right panel of Fig. 6, where we show the ROC curves for the adversarial autoencoder. As before, we evaluate the network on 10 independent test samples of 20,000 QCD jets and 20,000 top jets.

For the interplay between the mass de-correlation and the performance of the network the ROC curves are not the final word, though. Because the jet mass is removed from the autoencoder, we now see a clear top mass peak in the least QCD-like selection. This peak can be extracted using a shape analysis of the jet mass distribution with fully controlled side bands. This feature makes a huge experimental difference and clearly shows how the adversary in the jet mass promotes the autoencoder to a powerful experimental discriminator.

Finally, we can combine the same adversary part of the network with the 4-vector-based autoencoder described in Sec. 2.2. The combined loss function is now given by Eq. 8, but including the 4-vector-based loss function of Eq.(6). The ROC curve for a background shaping similar to the choice $\lambda = 5 \cdot 10^{-4}$ for the images shows that in the LoLa setup it is much

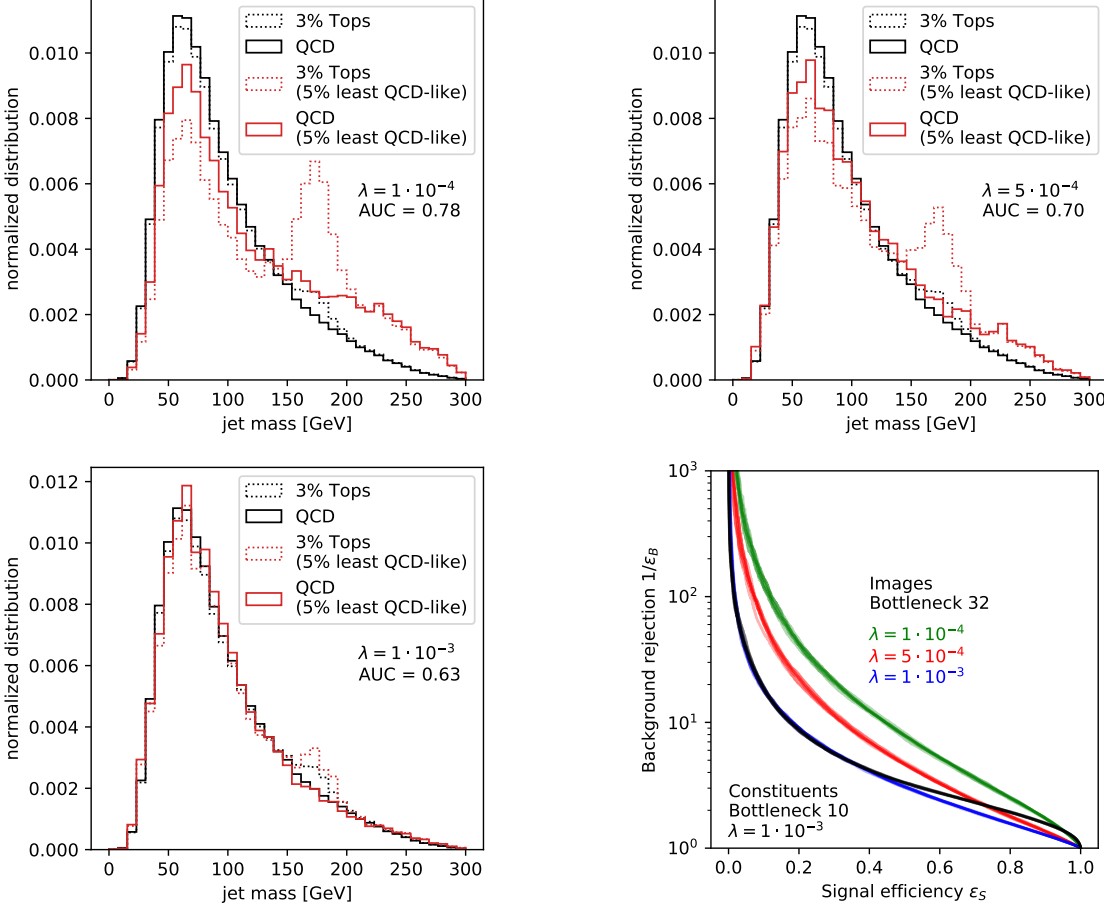

Figure 6: First three panels: jet mass distributions from the adversarial autoencoder with different values for $\lambda$, trained on pure QCD, and tested on pure QCD and a sample with 3% top jets. Lower right: ROC curves for the image-based and 4-vector-based adversarial autoencoders. The widths of the lines show the variation based on ten independent test samples for fixed training.

harder to de-correlate the jet mass. Correspondingly, the networks are less stable and have a worse performance. This is because the LoLa architecture in Eq.(5) focuses the network on learning the jet mass, which should then not be the one observable we de-correlate through the adversarial network. For that reason, we will focus on image-based adversarial autoencoders for the rest of this paper.

## 2.4 Realistic analysis setup

The problem in an actual analysis based on fully un-supervised learning on QCD jets will be that we cannot avoid a certain signal contamination of the training data. If the QCD training sample includes a small fraction of non-QCD, or in our case top jet, the autoencoder will accommodate top jets as QCD-like more easily. With the adversarial autoencoder we have developed an approach that can identify anomalous jets uncorrelated from any variable of choice.

In Fig. 7 we show the usual jet mass distribution for the image-based network, but trained on a QCD sample contaminated by 3% top jets. We keep $\lambda = 5 \cdot 10^{-4}$, but choose a much smaller bottleneck of 10 because the network now tends to accept tops as QCD-like jets, so we need to squeeze it harder in extracting non-QCD features. The performance with these

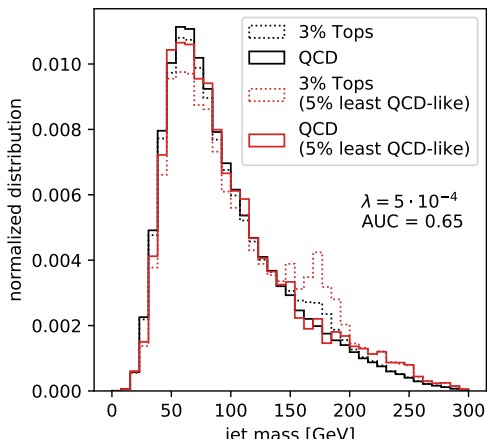

Figure 7: Jet mass distributions from the adversarial autoencoder trained on a mixed sample with 3% top jets using $\lambda = 5 \cdot 10^{-4}$, and tested on a QCD sample or on the same mixed sample.

settings is almost the same as for the adversarial training on QCD jets only, shown in Fig. 6, with an AUC of 0.65 instead of 0.70. This loss in performance can largely be recovered by a small change in the Lagrangian multiplier to $\lambda = 3 \cdot 10^{-4}$. Also the remaining background shaping in Fig. 7 is similar to the pure QCD case shown in Fig. 6. As hoped for, the top jets in the test sample are still collected as part of the least QCD-like jets, and they retain a distinctive mass peak which the squeezed adversarial network does not flatten.

This behavior opens the door to new strategies searching for physics beyond the Standard Model. We briefly sketch the application to a bump hunt using the invariant jet mass [58]. First, we define a region of phase space and an analysis variable. While our method can be applied to the inclusive QCD jet distribution, we focus on a search in jets with large transverse momentum as motivated by possible signals for hadronic decays of massive particles. Given a phase space region we use simulated QCD jets to set the hyper-parameters of the adversarial network, including the Lagrange multiplier. The three figures of merit are: flatness of the mass response, ability to identify a benchmark signal, and stability of the training.

Crucially, the actual training of the network already uses data from the same sample as we want to analyze. This means that we split the full data sample into statistically independent training and analysis samples. The key distribution is the jet mass for increasingly anomalous jets. It can be evaluated using standard bump-hunting techniques to extract a new physics signal. The signal jets can then be further dissected using orthogonal analysis techniques.

Because the training and the search rely on data in the same phase space region, the usually leading systematics do not enter. The remaining key uncertainty is the propensity of the network to induce a fake bump despite adversarial training. It can be reduced through a proper tuning of the hyper-parameters on simulation and verified using additional control regions in data. In case we see no signal, the network response can be used to set exclusion limits for arbitrary signal models. Compared to usual new physics searches the tables are turned: instead of training the network on simulation and applying it to data, we now train the autoencoder on data and apply it to simulation. In turn, the related systematic uncertainties have to be considered for exclusion limits.

# 3 Exotics in jets

While top decay jets are a great tool to test and benchmark our autoencoder, they are clearly not the most attractive application as the top is a known particle. Instead, we need to show how the autoencoder works in extracting other, exotic jets from a QCD sample where the parameters might not a priori be known. We will rely on two examples for this purpose: first we will test the autoencoder on a sample which includes a Higgs-like scalar decaying to four jets. It replaces the second, $W$-mass handle in the top jet by an increase in the subjet multiplicity. Second, we will use a modified, dark shower with QCD radiation as well as dark radiation off heavy dark quarks. The dark radiation produces missing energy and modifies the jet mass distribution, while leaving two hard jets with anomalous radiation patterns. For both of these models we show how the autoencoder with and without adversary can be used for a signal-independent LHC search.

## 3.1 Scalar decay to jets

As an alternative to the massive top jets we study a toy model with a Higgs-like scalar decaying to four charm jets through two light pseudoscalars,

$$pp \to (\phi \to aa \to c\bar{c}\, c\bar{c}) + \text{jets} . \tag{9}$$

The particle masses are $m_\phi = m_t = 175$ GeV and $m_a = 4$ GeV. We are not concerned with constraints on this toy model and choose the scalar mass such that we can easily compare our results with the top jet case and the pseudoscalar mass such that it decays to, for example, charm jets.

The light pseudoscalar are will be strongly boosted, and its decays should lead to four jets without a strong hierarchy in energy and without a distinctive mass scale aside from the jet mass. We simulate the signal with PYTHIA8.2.30 [60] and DELPHES3.3.3 [61], as usual ignoring multi-parton interaction and pile-up. The fat jets are anti-$k_T$ jets [66] with size $R = 0.8$, defined by FASTJET3.2.2 [67,68] with

$$p_{T,j} = 475 \dots 525 \text{ GeV} . \tag{10}$$

As before, the objects of the subjet analysis are particle flow objects [69] from the DELPHES E-flow. The leptons from the charm decays are taken into account for the calorimeter. For the pre-processing we center the jets in the $k_T$-weighted centroid before pixelization and use a range of $-0.75 \dots 0.75$ for the azimuthal angle and for the rapidity.

In Fig. 8 we show the main physics patterns of the scalar decay jets compared to the QCD background. In the left panel we see the number of constituents. Comparing to Fig. 1 we see that the general patterns are very similar, with the color-charged top leading to a slightly larger number of constituents. In the right panel we show the jet masses for the signal and the background. Both plots indicate that the heavy scalar signal is very similar to the top signal, but without the intermediate mass drop from the $W$-decay. This will force the de-correlated network to discriminate signal and background just based on the number of properties of the constituents from the scalar decay vs QCD radiation. In principle, we could increase the reach for this model by applying $c$-tagging, but for our toy model we explicitly do not want to use this additional information.

The setup of the autoencoder network with and without adversary is exactly the same as for the top case, including a bottleneck size of 32 units. The total size of the generated sample was 800,000 jets for training, and $\approx 250,000$ jets each for validation and final testing. In the left panel of Fig. 9 we include a ROC curve for the image-based autoencoder network without

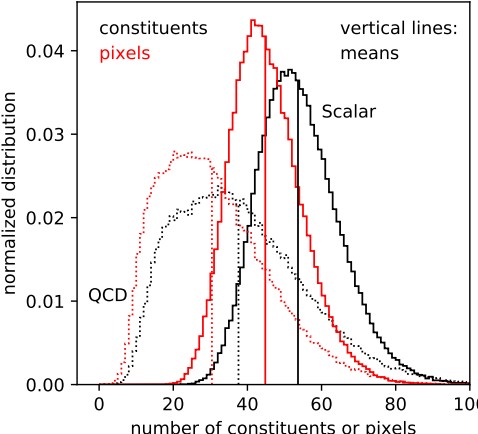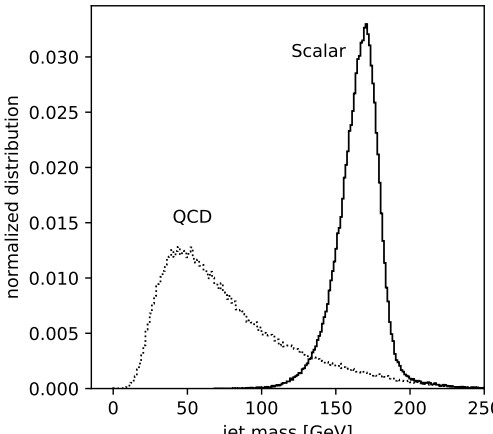

Figure 8: Left: numbers of constituents and of non-zero pixels for scalar decay jets and QCD. Right: truth-level jet mass distributions for the signal and the QCD background.

adversary, trained on QCD jets only. It corresponds to an AUC value of 0.90, comparable to the top case. As before, we can add an adversary to the autoencoder, to remove the information on the jet mass from the network and to generate control samples. This leads to a weaker performance of the network. For the same bottleneck of 32 units and a Lagrangian multiplier $\lambda = 10^{-3}$ we find the ROC curve given in Fig. 9 with an AUC value of 0.60. As mentioned before, this is significantly worse than for the top case, because the scalar is missing a second mass drop at intermediate masses.

To see the effect of the adversarial, we show the performance after training on pure QCD jets and evaluated on a sample including 3% signal jets in analogy to Fig. 6. Two sets of curves include all jets or the 5% least QCD-like jets in the right panel of Fig. 9. First, we indeed observe a small enhancement around $m_j = m_t$. While for our choice of the Lagrangian multiplier there remains a small background shaping, we also observe a clear signal enhancement for the least QCD-like events. However, the scalar example also shows the limitations of a subjet analysis where we cannot apply a mass drop and have to rely on difference similar to quark-gluon discrimination.

## 3.2 Dark showers

We use modified, dark showers [80, 81] as another benchmark scenario, independent of their new physics motivation through hidden valley models [82]. We assume that the model includes a heavy dark quark $q_v$ which can be pair-produced at the LHC. It undergoes showering in the dark and SM sectors and eventually decays to its SM-partner and a light dark boson, $b_v$, which is uncharged under all SM-gauge groups. This dark boson hadronizes into scalar and pseudoscalar dark meson states, collectively labelled as $\pi_v$ and assumed to have identical masses $m_{\pi_v} = 2m_{b_v}$. Depending on the model parameters the dark mesons can decay back to SM particles via a reverse of the production process, or leave the detector unobserved. The visible signature is therefore di-jets plus a variable amount of missing energy

$$pp \rightarrow q_v \bar{q}_v \rightarrow q\bar{q} + \not{E}_T \ . \tag{11}$$

However, the exotic production mechanism through a heavy color-charged dark quark leads to a sizeable amount of QCD radiation together with the dominant jets. It generates a jet mass spectrum with an upper edge at the dark quark mass. For our study we use a range of

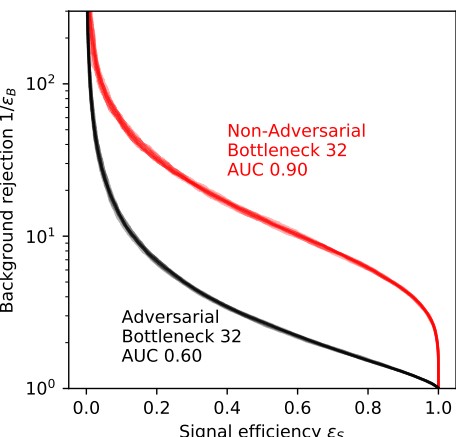 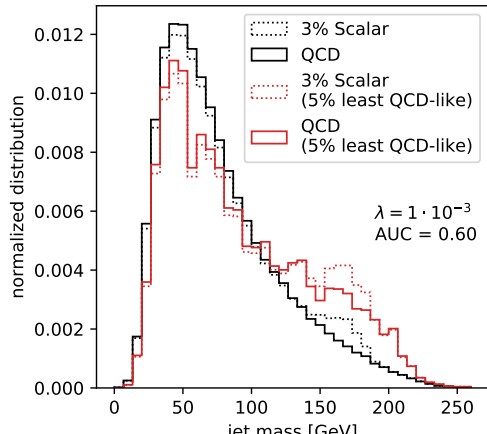

Figure 9: Left: ROC curves for the image-based autoencoders with and without adversary. Right: jet mass distributions from the adversarial autoencoder trained on pure QCD.

dark quark and dark boson masses. The dark gauge interaction we consider is $SU(3)_v$ with $\alpha_v = 0.1$, which is a PYTHIA default model.

The generation setup of for the dark showers is the same as for the heavy scalar in Sec. 3.1, only with a slightly higher $p_T$ range,

$$p_{T,j} = 575 \dots 625 \text{ GeV} . \tag{12}$$

The image preprocessing is identical to the scalar case with minimal pre-processing before pixelization.

For the dark shower model parameters we again ignore current experimental constraints and choose scenarios which best test and illustrate the behavior of our adversarial autoencoder. Similar to the top and heavy scalar cases we use a dark quark with mass $m_{q_v} = 200$ GeV. For a small meson mass of $m_{\pi_v} = 10$ GeV we see in the upper panels of Fig. 10 that the number of constituents and the jet mass are similar to the other new physics scenarios in the paper. In addition, we choose a more mass-degenerate case of $m_{q_v} = 200$ GeV and $m_{\pi_v} = 100$ GeV to test what happens in the absence of a peak in the jet mass altogether. For the dark shower samples we used 800000 training, 260000 validation and 280000 testing samples.

In the lower panels of Fig. 10 we first show the performance of the autoencoder without adversary. For both models we find excellent performance with AUC values of 0.78 ... 0.79. In the direct comparison, the autoencoder can more easily reject the peaked jet mass distribution, but at high efficiencies it is hard to separate the low-mass peak from QCD. For the adversarial network with $\lambda = 10^{-2}$ we now use 50 jet mass bins instead of the 10 used before. We find that the performance drops to a level comparable with the heavy scalar case with AUC value around 0.6, but with better jet mass de-correlation. As expected, the mass peak for the 5% least QCD-like events is broader and less pronounced for the mass-degenerate model. As for the scalar case, we clearly see that the autoencoder strategy works, but also that most of the relevant information is included in the jet mass distribution. In return, de-correlating this key observable for background control leads to a significant drop in performance.



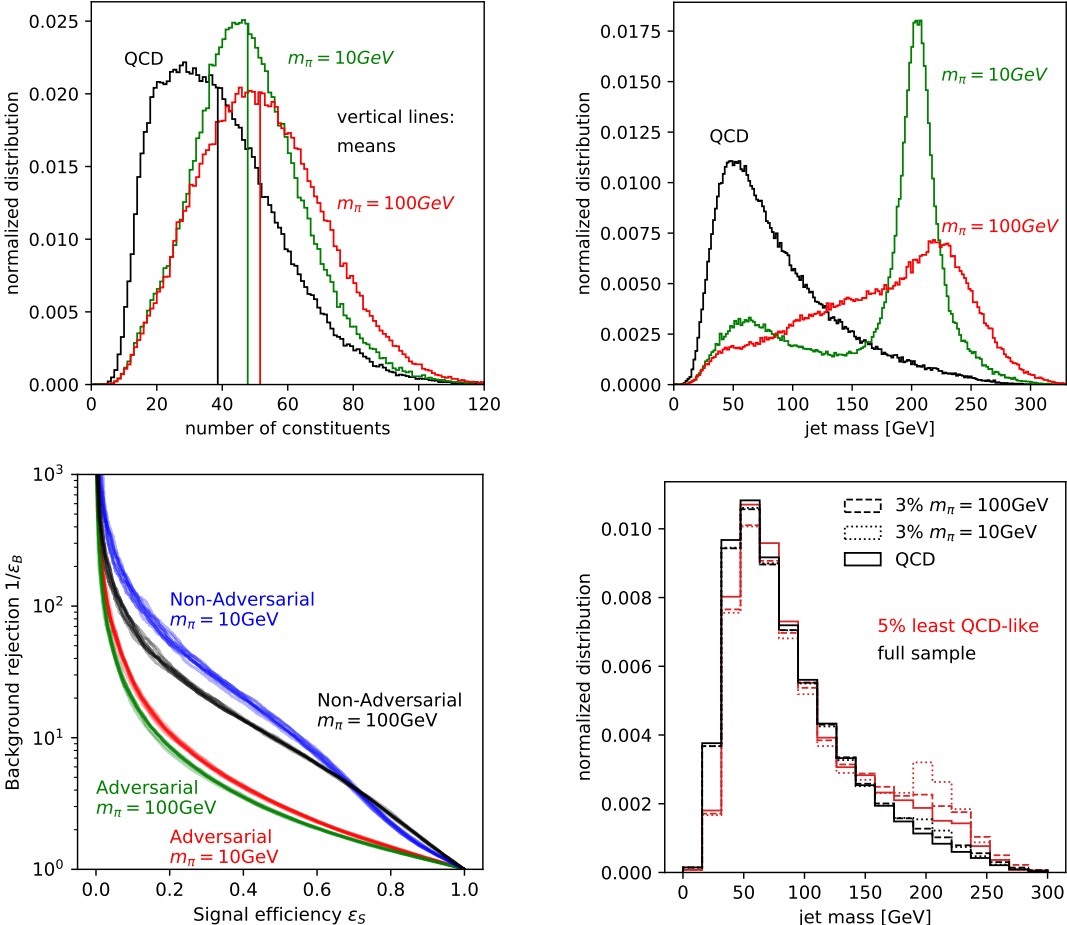

Figure 10: Autoencoder applied to a set of dark shower signals. Upper left: numbers of constituents for the dark shower models. Upper right: truth-level jet mass distributions for the different models. Lower left: ROC curves for the autoencoders with and without adversary. Lower right: jet mass distributions from the adversarial autoencoder trained on pure QCD.

## 4 Outlook

Anomalies in jets at the LHC can be extracted with the help of an autoencoder, a neural network based on low-level data and trained on QCD or other background samples only. We have shown that such a network extracts boosted hadronic top decays based on jet images or based on 4-vectors with a simplified LoLa structure. This technique is also compatible with other jet representations and network architectures. Its reduced performance as compared to specialized taggers is balanced by reduced systematic uncertainties in the absence of a well-defined signal model. Moreover, one autoencoder network realizing un-supervised learning for a given phase space region can be used to search for many different signals at the same time.

To further reduce experimental systematics, we propose to train and use an autoencoder network in the same phase space region. This requires full control of the background shaping. We extend our approach to an adversarial autoencoder based on jet images, de-correlating for example the jet mass from the training. This allows us to sort a jet sample by the loss function describing how QCD-like the jet is. We find (essentially) the same jet mass distribution for each slice in the loss function. For instance top decay jets are now collected in the least QCD-like

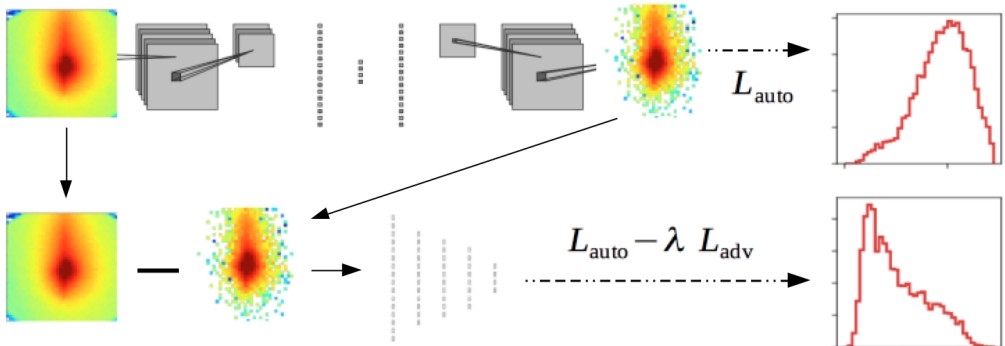

Figure 11: Illustration of our network setup.

slices and lead to a distinct peak in the jet mass.

Next, we have shown how to train the adversarial autoencoder on data with a signal contamination. In that case we typically make the autoencoder more restrictive and still find that the top jets are classified as the least QCD-like jets. We can still select them based on the network output and search for their distinctive peak in the jet mass distribution for non-QCD slices.

Finally, we have shown how the (adversarial) autoencoder can be used to not only extract top decay jets, but also decays of a heavy scalar to four quarks, or dark showers. Both of these models are significantly harder to extract than tops at the LHC. After de-correlating the jet mass, the different signals retain different amounts of information, allowing us to separate them from the QCD background. Given the universal structure of the autoencoder network this means that the experimental LHC collaborations could make their networks, trained on data, public and allow external groups to test if specific models would indeed be flagged as anomalies and are hence excluded.

While finishing this paper we heard of a similar, independent study, which is published in parallel to our work [83].

## Acknowledgments

We would like to thank David Shih and his group for the very friendly and constructive coordination. We are grateful to Michel Luchmann for help with the improved image pre-processing. Finally, we would like to thank the BOOST conference series for the encouraging atmosphere, without which papers like this might never be written.

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
