# Peer review of "QCD or What?"

_SciPost Physics, doi:SciPost Phys. 6, 030 (2019)_

## Round 2 · Referee Report · Anonymous · 2018-10-15

Strengths

Puts forward an innovative new technique for a high priority new area - search for BSM effects in jet substructure at the LHC. Makes a convincing attempt to reduce the model dependence of such an approach.

Weaknesses

1 - the model neglects underlying event and pile up effects, which are typically reduced by "grooming" jets using one or another technique. The authors do not comment or show whether their method works on groomed jets, but this would be relatively simple to do using the tools they have at hand, I think.

Report

I think this should be accepted, if my questions can be addressed/answered.

Requested changes

1 - show the MC statistics are high enough to support the conclusions, or generate more
2 - show or discuss how well the method should work on groomed (pile-up suppressed) jets.
3 - show or discuss impact of the detector simulation used
4 - address questions/requests for clarification in the attached PDF (which include the above as the most significant) (I also so highlighted some bits of text which look like typos or may need rephrasing)

Attachment

---

## Round 2 · Referee Report · Anonymous · 2018-11-5

Strengths

This paper takes a novel approach to a well known and difficult problem in jet physics, the identification of signal jets from QCD backgrounds. It does so by applying recent Machine Learning tools repurposed to work as (anti-)taggers.

Weaknesses

1. There are a few typos/missing words, and the paper would benefit from careful proofreading
2. The paper does not consider the robustness of the results to e.g. non-perturbative effects.
3. The loss functions used suffer from some drawbacks, such as sensitivity to rotations or soft and collinear splittings, and the authors do not discuss the impact of these limitations.

Report

I recommend this article for publication after minor changes are made to address my points below.

Requested changes

1- On page 2, when saying "we can choose our input format to deep learning analysis tools", do the authors mean here choose an input format best adapted for deep learning frameworks?
2- On page 2, regarding the use of jet images: it seems to me that while jet images have historically been the first representation used in conjunction with deep learning networks, there is no particular consensus on which input type is preferred, and in fact there has been substantial work in exploring other techniques. I would suggest citing some of these other methods in this paragraph as well, such as:
* arXiv:1702.00748
* arXiv:1704.02124
* arXiv:1704.08249
* arXiv:1710.01305
* arXiv:1712.07124
* arXiv:1807.04758
* arXiv:1810.05165
3- On page 2, regarding how to address systematic uncertainties: While I agree that this article presents an interesting angle, using adversarial networks to study some of these limitations, I think the statement is too broad. There are certainly other systematic uncertainties beyond those considered here.
4- On page 5, equation (2). An obvious downside to this loss function, and to the jet image approach in general, is that it is very sensitive to rotations: a small rotation, while leaving the physical properties mostly unchanged, will lead to a large value of the loss function. A discussion of this point and whether the authors have any insights into how it impacts the results would be useful.
5- On page 6, equation (3). The $(k_{\mu,i})$ matrix is not IRC safe: for example, a collinear splitting will result in a reshuffle of the columns, as well as a change of the values in eight of the entries. Did the authors study the impact of this unsafety?
6- On page 7, equation (7). Since the matrix compared before and after autoencoding can change substantially due to effects that are not physically relevant, e.g. soft or collinear splittings, does this impact the performance of the loss function?
7- On page 8. It would be interesting to see this study done on groomed jets, to remove the impact of soft wide angle partons on the jet mass considered as input.
8- On page 8, just after the middle of the page: "We know from many studies that the jet mass is the single most powerful observable in separating QCD jets from hadronically decaying heavy states". This is only true at parton level, without considering non-perturbative or pile-up effects. Otherwise, some of the many studies should be cited.
9- On page 12, the last sentence of section 2 is missing a "to".
10- On page 16, the second sentence of the second paragraph in the Outlook section is missing an "of".

---

## Round 2 · Referee Report · Anonymous · 2018-11-12

Strengths

(1) the authors demonstrate that adversarial networks can be used to decorrelate kinematic information when using machine-learning algorithms. The net result is that observables can be produced that unbiased by the application of the machine-learning algorithm. Such an unbiased observable would be useful in an experimental analysis. The authors demonstrate clearly that the kinematic observable that is decorrelated from the output variable of the adversarial network.

(2) the authors propose that weak supervision allows the adversarial network to be trained directly on data (instead of MC simulation) and in exactly the same phase space as the final search analysis. By doing this, they attempt to reduce (or entirely remove) experimental and theoretical systematic uncertainties that are present in the current searches, where machine-learning algorithms are trained on MC simulation. The paper demonstrates that the weakly -supervised adversarial networks can correctly classify signal and background, for injection of 3% signal of a variety of signal models (hadronic decays of tops, scalars or dark showers). The performance is reasonable, with an understandable reduction in signal/background separation when compared to non-adversarial networks, which is perhaps a price worth paying if systematic uncertainties are reduced.

Weaknesses

(1) The paper does not discuss the issue of imperfectly calibrated or badly-measured input objects. Experimentally, the inputs to the adversarial network (calorimeter clusters, tracks, particle-flow objects) are imperfectly calibrated, with sudden changes in the calibration at specific values of object transverse momentum and pseudo-rapidity. Furthermore, objects can be badly measured, due to non-Gaussian tails in the calorimeter resolution and kinks of charged particle tracks due to interactions with the material of the tracking detectors. Both of these effects could sculpt the invariant mass spectrum into a bump. What is not clear is how the weakly-supervised adversarial network would respond to these mis-measured jets (and it is unlikely that DELPHES produces such events). If the events show up in the 5% of events that are ‘least QCD-like’, they would likely be interpreted as a signal. The paper would benefit from a discussion about experimental effects such as these. Ideally, a miscalibration could be injected into the simulation for quantitative studies.

Report

The authors propose that weakly-supervised adversarial networks can be used to address known issues that arise when using machine learning algorithms to search for signatures of New Physics in boosted hadronic jets. The idea is a good one and the approach is validated with simulated data. However, more discussion and/or tests are needed regarding the experimental effects of imperfect calibration and badly measured input objects.

Requested changes

(1) The authors should add a discussion about experimental effects such as imperfect calibration and badly measured input objects

(2) The authors should directly test the impact of imperfect calibration and badly measured input objects by injecting a known miscalibration to the particle-flow objects to see the impact on the weakly-supervised adversarial network. It would build confidence if such effects were shown to be negligible or could be mitigated in some fashion.

---

## Round 3 · Referee Report · Jonathan Butterworth · 2019-1-23

Report

My only remaining query is the question I had about constituent masses, where I think we may be misunderstanding each other. The reply is simply that the constituent mass it E^2-p^2. However, if the constituents are charged particle tracks we know p, if they are calorimeter clusters we know E. In a sophisticated particle-flow reconstruction we have a measure of both. But the mass of the constituent particles is not so well known in any case, and I can image that once the (large) boost of the jet is removed by boosting the particles into the rest frame of the fat jet, the mass of the constituents has a significant impact. So: are the true particle masses used? Or are all constituents taken or be massless? Or something else? Or can you convince me it is irrelevant or I have misunderstood?

Requested changes

see above, please clarify.

  • validity: -
  • significance: -
  • originality: -
  • clarity: -
  • formatting: -
  • grammar: -

Author:  Jennifer Thompson  on 2019-01-30  [id 418]

(in reply to Report 1 by Jonathan Butterworth on 2019-01-23)
Category:
answer to question

Thank you for the clarification. We have produced distributions (attached) for QCD and top jets for both the maximum constituent mass/jet mass and the maximum constituent mass/energy for 10000 jets. In all cases the constituent mass is much smaller than both the constituent energy and the jet mass. The constituent masses are therefore insignificant in these samples, even after boosting to the jet rest frame.

Attachment:

constituent_masses.pdf

Anonymous on 2019-02-03  [id 424]

(in reply to Jennifer Thompson on 2019-01-30 [id 418])
Category:
remark

Thanks, that seems clear enough!

---

## Round 3 · Referee Report · Anonymous · 2019-1-31

Report

In my previous report, I suggested two additions:

(1) The authors should add a discussion about experimental effects such
as imperfect calibration and badly measured input objects

(2) The authors should directly test the impact of imperfect calibration
and badly measured input objects by injecting a known miscalibration to
the particle-flow objects to see the impact on the weakly-supervised
adversarial network. It would build confidence if such effects were
shown to be negligible or could be mitigated in some fashion.

The authors reply was:

* Autoencoding is not a weakly supervised but an unsupervised technique. The final discriminant is trained and evaluated on data. Therefor no effects due to calibration or bad measurements exist.

My follow up:

I appreciate the fact that this unsupervised algorithm is trained directly on data, but that does not mean there is no impact of imperfect detector calibration. In fact, I believe that unsupervised algorithms are currently being investigated by the LHC collaborations precisely to find detector issues that are not known about. Let me outline in detail the problem below.

The response of calorimeters to particles is non-linear and highly dependent on the detector geometry. This response is corrected (either at calorimeter or jet level) using test-beam data and MC simulations. The correction cannot be perfect. Furthermore, the data can contain noise and/or hot-cells that have not been correctly identified and removed from the object reconstruction.

The upshot of experimental reality above is that the unsupervised algorithm can learn features of the detector rather than the underlying physics. It is not clear to what extent this affects the current proposal. I would very much like to know the answer by quantitative tests, but I understand this might be beyond the scope of the current study (which is interesting and should be published). However, a short discussion on experimental effects would be useful to the reader and to promote further investigation.

Requested changes

The authors should add a discussion regarding the fact that unsupervised algorithms could learn features of a imperfectly calibrated (or noisy) detector and that additional checks using control regions in the data would be needed to make sure that any such experimental issues were not mistaken for a new physics signal.

  • validity: -
  • significance: -
  • originality: -
  • clarity: -
  • formatting: -
  • grammar: -

Author:  Jennifer Thompson  on 2019-02-01  [id 423]

(in reply to Report 2 on 2019-01-31)
Category:
answer to question

We agree that this is an important point for us to address, and would like to add the following paragraph to the paper:

"A set of events flagged by the autoencoder as anomalies does not automatically qualify as a signal of new physics. It is standard experimental procedure to test whether any signal could be caused by detector effects. Typical tests include checks whether events cluster geometrically (all jets originate from a specific region in the $\eta$-$\phi$ plane, hinting at a misbehaving region of the calorimeter) or temporally (from a specific run or run-period, hinting at problematic LHC or detector conditions). In the case of autoencoding jet images, an additional test would be an analysis of the correlation with well-understood substructure variables such as n-subjettiness, which is opportune to understand the topology of the identified signal. Finally, mis-calibrations of the jet-energy that cause an artificial mass peak can be taken care of using control regions --- if the mass peak is present in sidebands as well, it is likely a miscalibration. All of these are relevant experimental considerations and should be included in any concrete study. However, autoencoding is no more susceptible (and arguably less so) than traditional techniques based on MC simulation."

Anonymous on 2019-02-05  [id 428]

(in reply to Jennifer Thompson on 2019-02-01 [id 423])

That looks good to me. Thanks for adding this paragraph. I have no further comments/requests.

---

## Round 3 · Referee Report · Anonymous · 2019-2-11

Report

I am satisfied with the changes provided by the authors and recommend the paper for publication.
A few minor points (referring to Review 2)

2-
I would suggest citing all the different ML methods on page 1, as the authors wrote in their reply, but which does not seem to be the case in the v3: 1710.01305, 1712.07124, 1807.04758 are still only cited in section 2.3.

4-
My point here is that this behaviour of the loss function will lead to undesirable properties. For example, for two particles that have very similar energies but end up in neighbouring bins, the loss function will be large even though the jets are almost identical.

6-
The issue is that a jet where one parton undergoes a soft or collinear splitting is identical to the one without that splitting. Therefore, an ideal autoencoder should in principle be able to collapse these two example jets into the same latent space projection, which can not be the case here because the loss function does not consider them equivalent. But I agree that this problem is perhaps beyond the scope of this paper.

---

## Round 3 · List of Changes

Review 1

1 - show the MC statistics are high enough to support the conclusions,
or generate more

*Added event counts also to dark shower section. We used more than 100k events for each signal and background for the final testing. This means even the 5% least QCD like histogram still has more than 5000 events.

2 - show or discuss how well the method should work on groomed (pile-up
suppressed) jets.

*We now explicitly recommend using PU removal techniques that do not rely on grooming.

3 - show or discuss impact of the detector simulation used

*Similarly, no detailed detector simulation was included. We expect the autoencoder to learn novel jet-shape variables from the distributions of constituents. There is no a-priori reason why these jet shapes would suffer from larger effects due to the detector simulation than widely used variables like groomed mass, n-subjettiness or energy correlation functions. For the practical application of the autoencoder we foresee training on data, making this technique even less subject to differences between data and simulation than ordinary approaches.

4 - address questions/requests for clarification in the attached PDF
(which include the above as the most significant) (I also so highlighted
some bits of text which look like typos or may need rephrasing)
+pdf comments: 1808.08979v2_report_attachment-1.pdf (attached)
Typos fixed:
-> page 5: Statically -> Statistically
-> page 12: added commas: to, for example,
-> page 12: as usually -> as usual
-> page 14: changed order of references 39<->40
-> page 16: added 'to thank' twice to the acknowledgments

page 3: encodes/preserves instead of 'bottleneck
describes the features'.

*Changed to encodes

page 3: Changed 'calorimeter information' to 'pixellated energy'

page 3: Comment on reliability DELPHES as a detector simulator. Is pixellation the dominant effect?

*Delphes is now the de-facto standard for this kind of phenomenological study, especially for jet substructure. The main effects are indeed a semi-realistic granularization and smearing of the energy.

page 4: how does this work with groomed jets?

*There is no fundamental reason why this should not work for groomed jets as well. We however do not suggest applying grooming together with the autoencoder as grooming algorithms make inherent assumptions on the structure of a parton shower and a major feature of the autoencoder is that it does not need these assumptions. Added a statement along these lines to the text.

page 6: what is used for constituent masses?

*The masses of the constituents are not used for boosting, the 4-vector is directly boosted into the jet rest frame, so effectively the constituent mass is E^2-p^2.

Page 6: why is LoLa needed when the information is already included in the 4-vector?

*The LoLa implementation transforms the 4-vector such that it includes physically relevant information. This helps the network to learn physics and removed symmetries in the phi/eta plane which the network would otherwise have to learn itself. This implementation is discussed further in the reference: arXiv:1707.08966 [hep-ph].

Page 8: the question to the data is

*Rephrased

page 9: Comment jargon-heaviness.

*We would like to have a couple of sentences detailing the technical implementation for anyone interested.

page 12: How do you know if an analysis technique is orthogonal?

*The anomalous events can be analysed further in multiple ways. One example of an orthogonal feature would be the Njet distribution of the full event, as this is a jet-by-jet tagger

page 13: display mean on number of constituents plots (for comparison)

*Added plots with means

Page 16:
"We find (essentially) the same flat jet mass distribution for each slice in the loss function. For instance top decay jets are now collected in the least QCD-like slices and lead to a distinct peak in the jet mass"

*Removed the word flat

Review 2
1- On page 2, when saying "we can choose our input format to deep
learning analysis tools", do the authors mean here choose an input
format best adapted for deep learning frameworks?

*Added 'This allows us to pick the data format that is best suited to a given problem.'

2- On page 2, regarding the use of jet images: it seems to me that while
jet images have historically been the first representation used in
conjunction with deep learning networks, there is no particular
consensus on which input type is preferred, and in fact there has been
substantial work in exploring other techniques. I would suggest citing
some of these other methods in this paragraph as well, such as:
* arXiv:1702.00748 (cited)
* arXiv:1704.02124 (cited)
* arXiv:1704.08249 (cited)
* arXiv:1710.01305 (cited)
* arXiv:1712.07124 (cited)
* arXiv:1807.04758 (cited)
* arXiv:1810.05165 (added)

*Now all the approaches are cited on page 1. Added a comment in the conclusions that "This technique is also compatible with other jet representations
and network architectures." We only tested images and LoLa because of prior experience, but we currently see no limitation in using the autoencoder on other representations.

3- On page 2, regarding how to address systematic uncertainties: While I
agree that this article presents an interesting angle, using adversarial
networks to study some of these limitations, I think the statement is
too broad. There are certainly other systematic uncertainties beyond
those considered here.

*Also added a statement on refiner networks. Uncertainties that arise from differences between data and simulation can in general be "solved" by training on data as proposed by the autoencoder. The adversarial training here is only used to control correlations, not to surpress systematic uncertainties.

4- On page 5, equation (2). An obvious downside to this loss function,
and to the jet image approach in general, is that it is very sensitive
to rotations: a small rotation, while leaving the physical properties
mostly unchanged, will lead to a large value of the loss function. A
discussion of this point and whether the authors have any insights into
how it impacts the results would be useful.

*It is true that jet images have problems with rotations. However our pre-processing rotates jet images to a standardised format. Secondly - it is even less of a problem for the loss function: The loss function measures the difference between the input image and the input image processed by the autoencoder. Unless the autoencoder LEARNS to rotate - and there are no training incentives for it to do so - there will be no relevant rotations.

5- On page 6, equation (3). The (kμ,i) matrix is not IRC safe: for
example, a collinear splitting will result in a reshuffle of the
columns, as well as a change of the values in eight of the entries. Did
the authors study the impact of this unsafety?

*The authors are aware that LoLa is inherently IRC unsafe. Again we would like to point out the on-going top tagging comparison and especially the EFN vs PFN result: Most of recent gains in performance through machine learning seem to be inherently IRC unsafe. This is a very interesting question for further study, but beyond the scope of this article. See also previous and next comment.

6- On page 7, equation (7). Since the matrix compared before and after
autoencoding can change substantially due to effects that are not
physically relevant, e.g. soft or collinear splittings, does this impact
the performance of the loss function?
similar to point 4

*No. The loss function is not comparing two "identical" jets where jet A underwent a soft splitting and jet B did not; the loss function compares the initial jet A with A'=decoder(encoder(A)) and B with B'. The autoencoder does not induce splittings/etc. So there is no place where these effects could occur. It is only relevant that the encoding/decoding procedure works equally well for A and B if they are physically similar, but this is ensured by large training statistic.

7- On page 8. It would be interesting to see this study done on groomed
jets, to remove the impact of soft wide angle partons on the jet mass
considered as input.
- copy of ref 1.2 -

8- On page 8, just after the middle of the page: "We know from many
studies that the jet mass is the single most powerful observable in
separating QCD jets from hadronically decaying heavy states". This is
only true at parton level, without considering non-perturbative or
pile-up effects. Otherwise, some of the many studies should be cited.

*Changed to "a powerful observable".

9- On page 12, the last sentence of section 2 is missing a "to".
*fixed

10- On page 16, the second sentence of the second paragraph in the
Outlook section is missing an "of".
* fixed

Review 3

(1) The authors should add a discussion about experimental effects such
as imperfect calibration and badly measured input objects

(2) The authors should directly test the impact of imperfect calibration
and badly measured input objects by injecting a known miscalibration to
the particle-flow objects to see the impact on the weakly-supervised
adversarial network. It would build confidence if such effects were
shown to be negligible or could be mitigated in some fashion.

* Autoencoding is not a weakly supervised but an unsupervised technique. The final discriminant is trained and evaluated on data. Therefor no effects due to calibration or bad measurements exist.

---

## Editorial Decision

published